# What Does Atypical Chronic Lymphocytic Leukemia Really Mean? A Retrospective Morphological and Immunophenotypic Study

**DOI:** 10.3390/cancers16020469

**Published:** 2024-01-22

**Authors:** Giovanni D’Arena, Candida Vitale, Giuseppe Pietrantuono, Oreste Villani, Giovanna Mansueto, Fiorella D’Auria, Teodora Statuto, Simona D’Agostino, Rosalaura Sabetta, Angela Tarasco, Idanna Innocenti, Francesco Autore, Alberto Fresa, Luciana Valvano, Annamaria Tomasso, Lorenzo Cafaro, Daniela Lamorte, Luca Laurenti

**Affiliations:** 1Immuno-Hematology and Transfusion Medicine Unit, “San Luca” Hospital, 84078 Vallo della Lucania, Italy; rosalaurasabetta@libero.it (R.S.); angelatarasco@uslnordovest.toscana.it (A.T.); 2A.O.U. Città della Salute e della Scienza di Torino and Department of Molecular Biotechnology and Health Sciences, Division of Hematology, University of Torino, 10125 Torino, Italy; candida.vitale@unito.it; 3Hematology and Stem Cell Transplantation Unit, Centro di Riferimento Oncologico della Basilicata (IRCCS-CROB), 85028 Rionero in Vulture, Italy; giuseppe.pietrantuono@crob.it (G.P.); oreste.villani@crob.it (O.V.); giovanna.mansueto@crob.it (G.M.); simona.dagostino@crob.it (S.D.); 4Laboratory of Clinical Pathology, Centro di Riferimento Oncologico della Basilicata (IRCCS-CROB), 85028 Rionero in Vulture, Italy; fiorella.dauria@crob.it; 5Laboratory of Clinical Research and Advanced Diagnostics, Centro di Riferimento Oncologico della Basilicata (IRCCS-CROB), 85028 Rionero in Vulture, Italy; teodora.statuto@crob.it (T.S.); valvano.luciana@gmail.com (L.V.); 6Hematology Unit, Fondazione Policlinico Universitario A. Gemelli IRCCS, 00168 Rome, Italy; idanna.innocenti@policlinicogemelli.it (I.I.); francesco.autore@policlinicogemelli.it (F.A.); alberto.fresa@guest.policlinicogemelli.it (A.F.); annamariatomasso2@gmail.com (A.T.); luca.laurenti@unicatt.it (L.L.); 7Immuno-Hematology and Transfusion Medicine Unit, “Immacolata” Hospital, 84073 Sapri, Italy; l.cafaro99@gmail.com; 8Laboratory of Preclinical and Translational Research, Centro di Riferimento Oncologico della Basilicata (IRCCS-CROB), 85028 Vulture, Italy; daniela.lamorte@crob.it

**Keywords:** chronic lymphocytic leukemia, atypical, immunophenotype, morphology, flow cytometry, prognosis

## Abstract

**Simple Summary:**

Chronic lymphocytic leukemia (CLL) may be atypical in terms of the cell morphology picture, but also with regard to the surface immunophenotypic profile. Aiming at assessing the impact of morphology and immunophenotype in defining the atypical characteristics of CLL in terms of clinical–biological features and prognosis, a retrospective analysis of a large cohort of CLL patients was performed. We found that morphology better predicts the prognosis of atypical CLL compared to immunophenotypic analysis. Also, discordant cases in terms of immunophenotype and morphology did not identify specific prognostic groups. Overall, the question that still needs to be answered is: does it make sense to focus on morphology and immunophenotypic features in CLL in the era of molecular markers used as prognostic indicators?

**Abstract:**

Atypical chronic lymphocytic leukemia (CLL) is still defined according to morphological criteria. However, deviance from the typical surface immunological profile suggests an atypical immunological-based CLL. A large cohort of patients with CLL was retrospectively evaluated aiming at assessing morphological (FAB criteria), immunophenotypical (two or more discordances from the typical profile), and clinical–biological features of atypical CLL. Compared to typical cases, morphologically atypical CLL showed a greater percentage of unmutated IgVH and CD38 positivity, and a higher expression of CD20. Immunophenotypically atypical CLL was characterized by more advanced clinical stages, higher expression of CD20, higher rate of FMC7, CD79b and CD49d positivity, and by an intermediate–high expression of membrane surface immunoglobulin, compared to typical cases. When patients were categorized based on immunophenotypic and morphologic concordance or discordance, no difference emerged. Finally, morphological features better discriminated patients’ prognosis in terms of time-to-first treatment, while concordant atypical cases showed overall a worse prognosis. Discordant cases by immunophenotype and/or morphology did not identify specific prognostic groups. Whether—in the era of molecular markers used as prognostic indicators—it does make sense to focus on morphology and immunophenotype features in CLL is still matter of debate needing further research.

## 1. Introduction

Chronic lymphocytic leukemia (CLL) is the most common form of leukemia in the Western world [1]. More than 5000/µL circulating clonal B-lymphocytes co-expressing CD5 and CD23 are required for the diagnosis according to the International Working Group on CLL (IwCLL) criteria [2]. The French–American–British (FAB) cooperative group diagnostic criteria, established on the morphologic features of peripheral blood lymphocytes, are still currently used to classify CLL into two main categories: typical (approximately 80% of cases) and atypical CLL [3]. In the former, more than 90% of lymphocytes are small-to-medium sized with relatively normal morphology and characteristically clumped chromatin patterns; in the atypical form (the so-called CLL, mixed cell types), two forms have been described: (i) a dimorphic picture of small lymphocytes and prolymphocytes (>10% and <55% of lymphocytes), designated CLL/PLL; (ii) a spectrum of small-to-large lymphocytes with occasional (<10%) prolymphocytes (Table 1 and Table 2) [3].

The immunological profile of CLL is now well defined: monoclonal B-cell lymphocytes co-expressing CD5 and CD23, with a low expression of CD20 and CD22, and low surface immunoglobulin (Ig) expression [4]. CD79b and FMC7 are also generally absent. On this basis, British investigators in the ‘90s first tried to establish immunophenotypic criteria to diagnose CLL and differentiate it from other neoplastic B-cell chronic lymphoproliferative disorders (B-CLD) often requiring a specific and different therapeutic approach [5,6]. A scoring system currently used was defined based only on immunophenotypic criteria [5,6]. Furthermore, more recently, CD200 and CD43 have been shown to be helpful in accurately identifying CLL [7,8,9]. In light of this, other scores based on the immunophenotype have been proposed to increase the diagnostic ability of the British score [7,8]. However, no well established and shared criteria have been proposed to define atypical CLL from an immunophenotypic point of view like the FAB morphological subclassification, despite some tentative attempts [10,11].

In this study, we reviewed data from our database selecting a cohort of patients with CLL diagnosed at our institutions aiming at evaluating morphological, immunophenotypic, and clinical–biological features of atypical CLL.

## 2. Materials and Methods

This study was performed according to the informed consent procedure approved by local internal Review Board (Protocol no. 20140040750—18 November 2014), and it conforms to the provisions of the Declaration of Helsinki. 

One hundred and fifty-three patients diagnosed with CLL at our institutions between February 2001 and December 2019, for whom peripheral blood films were stored at diagnosis, were enrolled in this study. 

Demographics, clinical, and biological features at study entry are reported in Table 3. 

The archived May–Grunwald-stained peripheral blood films were independently re-examined by three investigators (G.D., G.P., and O.V.) and scored for the final classification according to the FAB criteria (Table 1 and Table 2) [3]. At the first evaluation, a 97% agreement was obtained. Discordant cases were re-evaluated and discussed to obtain a definitive conclusion resulting in a final 100% agreement.

A panel of monoclonal antibodies conjugated with fluorochromes was used to study the complete surface immunophenotype of each sample: CD5, CD20, CD22, CD23, CD38, CD43, CD49d, CD79b, CD200, FMC7, and kappa and lambda light chains. The methodology used was detailed elsewhere [13]. Surface antigen expression density, as well as surface membrane Ig density, were expressed as the ratio between isotypic control and monoclonal antibodies or smIg mean channel. In Table 4, the immunologic markers used are summarized and also their significance for B cells and for B cell-related malignancy. Finally, to define atypical immunophenotypic CLL, we used the criteria reported in Table 5 and detailed in the legend and based on the published literature and our own experience (Figure 1) [9,10,11,14,15,16,17].

Immunophenotypically atypical CLL is defined as 2 or more (score ≥ 2) discordance from the typical immunophenotypic CLL profile (CD20 and smIg low density, FMC7 and CD79b negativity, and CD200 positivity.

Typical CLL immunophenotype: CD19+ CD5+ CD23+ CD20+(low intensity) CD43+ CD200+ CD79b− sIgkappa+(low intensity).

Atypical CLL immunophenotype: CD19+ CD5+ CD23+ CD20+(high intensity) CD43+ CD200+ CD79b+ sIgkappa+(high intensity). The arrows indicate the 3 discrepancies in respect of the typical CLL immunophenotype.

### Statistics

Descriptive statistics were used to summarize patients’ characteristics. For categorical variables, differences between groups were tested with chi-square test or Fisher’s exact test, whereas the t-test or Mann–Whitney test were applied to analyze continuous variables. Time-to-first treatment (TTFT) was defined as the time interval between the date of CLL diagnosis and the date of first treatment or last follow-up. TTFT was estimated using the Kaplan–Meier method, and differences between groups were evaluated with the log-rank test. Statistical analyses were performed using GraphPad Prism version 8. A *p* value < 0.05 was considered significant.

## 3. Results

As reported in Table 6, morphologically atypical CLL showed a greater percentage of unmutated IgVH and CD38 positivity, and a higher expression of CD20, compared to typical cases. Conversely, when patients were categorized according to the immunophenotypic profile, more advanced clinical stages, a higher rate of FMC7, CD79b and CD49d positivity, a higher CD20 expression, and more frequent intermediate–high smIg density expression were found in the atypical vs. typical cases. Moreover, CD43 was more frequently undetectable in the immunophenotypically atypical cases (Table 7). Finally, when patients were categorized into four groups, based on both immunophenotypical and morphological features (i.e., (1) immunophenotypically and morphologically typical, (2) immunophenotypically and morphologically atypical, (3) discordant: immunophenotypically typical/morphologically atypical, (4) discordant: morphologically typical/immunophenotypically atypical) no relevant differences emerged (Table 8). The time-to-first treatment (TTFT) was used as a prognostic surrogate (Figure 2A). The categorization according to morphological features discriminated a worse or better prognosis (Figure 2B), whereas the immunophenotypic profile did not (Figure 2C). In light of this, in the morphological atypical subgroup, more advanced stages of patients were found. Of interest, cytogenetic abnormalities detected by FISH were not differently distributed in patients’ groups, when both atypical morphological and immunophenotypic criteria were evaluated. Finally, concordant atypical CLL (immunophenotypically and morphologically atypical) showed a worse prognosis in terms of TTFT when the four groups of patients were separately analyzed (Figure 2D).

When patients were divided into four groups, the median time-to-first treatment (TTFT) was 130 months for the immunophenotypically and morphologically typical group, 33 months for the immunophenotypically and morphologically atypical group, 51 months for the discordant: immunophenotypically typical/morphologically atypical group, and 222 months for the discordant: morphologically typical/immunophenotypically atypical group (*p* = 0.0001) (Figure 2D).

## 4. Discussion

The FAB cooperative group firstly established CLL diagnostic criteria on the basis of morphologic features of peripheral blood lymphocytes. These criteria are still currently used to classify CLL into typical and atypical cases [3]. For a long time, only these criteria were used to define prognosis of the disease. However, discrepancies were found among investigators when cells were evaluated by microscopy, and differences in incidence of atypical forms occurred in different cohorts. Schwarz et al. in their report concluded that the discrepancy in the percentages of the morphologically atypical cases reported in their study and a Belgic study was caused by the subjective nature of reading the morphological slides (Table 9) [14,16]. This is probably the main reason why morphological evaluation of CLL cells is not regarded as a reliable tool, and a careful cytological evaluation of the peripheral blood smears of CLL is needed. In light of this, Marionneaux et al. used a digital imaging technology (Cellavision AB digital imaging system; Lund, Sweden) enabling cell identification due to the simultaneous display of cells on a high-definition wide screen monitor with a faster and more objective classification of lymphocyte variants [36]. In addition, this digital microscopy technology appeared to be feasible, rapid, and an inexpensive screening tool.

Since the 1990s, researchers have focused their attention on the prognostic significance of atypical CLL, evaluated not only on a morphological basis but also using immunophenotypic data, thanks to the emergence of diagnostic tools such as flow cytometry (Table 7) [5,10,11,14,15,16,17,18]. However, well defined immunophenotypic criteria have not been established to characterize an atypical CLL. Among others, Finn et al. classified as immunophenotypically atypical those CLL cases that deviated from the typical phenotype: bright CD20 positivity, bright surface light chain positivity, or absence of CD23 staining [10]. In the work of Ting et al., cases scoring four or five points of five were classified as CLL (11). In CD5+ cases which scored one point for CD5 positivity, if the total score was <4, samples were evaluated for evidence of a t(11;14) translocation. If there was no evidence of such a translocation by cytogenetics or FISH, or if cyclin D1 was negative with immunohistochemistry, the case was classified as atypical CLL. Of this latter, only seven cases existed, three had dim or negative CD23 expression, all had partial or total FMC7 expression, and five had moderate–bright surface light chains. CD200 was strongly expressed on monoclonal cells of all typical CLL and on all seven atypical CLL samples. The CD200 mean fluorescence intensity and percent of positive cells in the atypical CLL samples were similar to the typical CLL cases. Two other studies reported positive CD200 expression on all atypical CLL cases [37,38]. In a retrospective analysis of flow cytometry data used to assess the feasibility of a cell-based proteomic approach to FCM by unsupervised cluster analysis, Habib and Finn showed that 14 atypical CLLs (out of 81 patients with CLL) were skewed toward “atypical” CLL characterized by high CD20, CD22, FMC7, and light chain, and low CD23.

More recently, the emergence of molecular and genetics testing seems to have obscured the prognostic significance of atypical CLL. In light of this, whether it still make sense to evaluate the deviance from typical CLL morphological and/or immunophenotypic features in CLL today is an issue that needs to be better clarified. Discordant data have been published so far. While some authors have reported a poor prognosis of atypical CLL, others were not able to demonstrate this. Our data showed that in morphologically atypical CLL only a greater unmutated IgVH, CD20 at a higher density and CD38 expression were detected while, when patients were categorized according to immunophenotypic profile, more advanced clinical stages, more frequent FMC7, CD79b, CD49d, CD20 at high density, smIg at intermediate–high density expression were found and CD43 was more frequently undetectable. Whether the differences between morphologically typical and atypical cases in terms of IgVH mutational status and CD38 and CD20 expression might be an expression of a somehow different disease in terms of cell-of-origin unmutated IgVH cases more observed in atypical CLL could be the expression of the development of a different cell needs to be addressed by specific studies. Finally, when patients were categorized according to immunophenotype and morphology concordance or discordance no difference emerged. According to TTFT, the categorization by morphology better discriminated a worse or better prognosis, differently from the immunophenotypic profile. In our cohort, cytogenetic abnormalities detected by FISH were not found to be different when both atypical morphological and immunophenotypic criteria were evaluated. Finally, concordant atypical CLL showed again a worse prognosis when concordant and discordant cases were evaluated. 

## 5. Conclusions

Usually identified by morphology, atypical CLL has shown a poor prognosis. We performed a retrospective analysis of a large cohort of CLL patients followed at our institutions, trying to associate morphology, immunophenotype, and molecular markers with a better diagnostic and prognostic definition.

Taken together, our data showed that morphological atypical features identify a subgroup of patients with poorer prognosis as well as atypical cases for both immunophenotype and morphology. On the contrary, discordant cases by immunophenotype and/or morphology did not identify a specific prognostic subgroup. In addition, no differences were found in the distribution of cytogenetic abnormalities among typical, atypical, and immunophenotypically/morphologically discordant groups. Dedicated studies comparing morphology and immunophenotype are welcome, using stringent criteria to define morphological atypia to definitively define their role in the era of genetic and molecular markers.

## Figures and Tables

**Figure 1 cancers-16-00469-f001:**
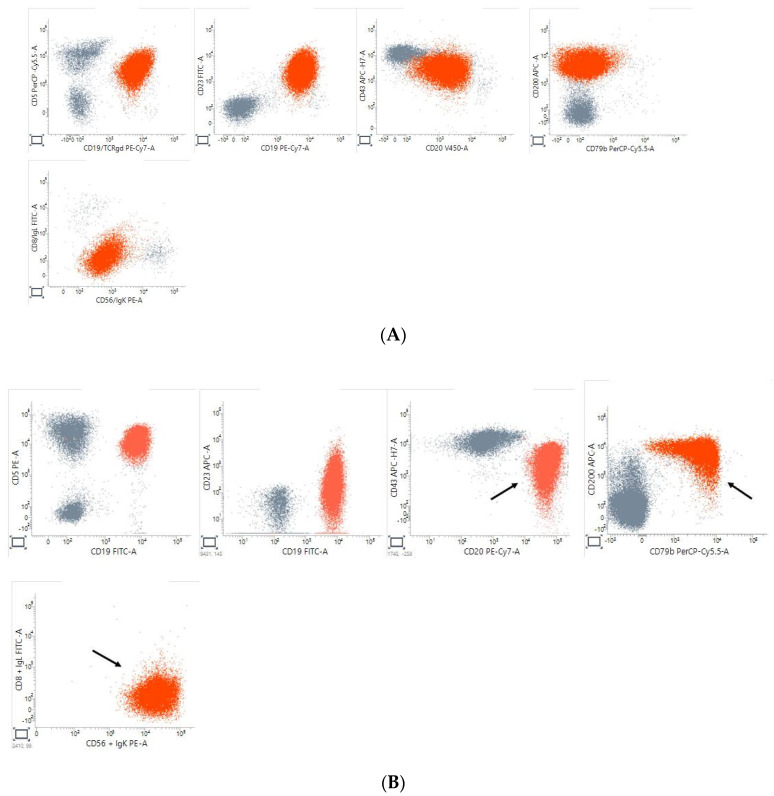
(**A**). Typical CLL immunophenotype. (**B**). Atypical CLL immunophenotype.

**Figure 2 cancers-16-00469-f002:**
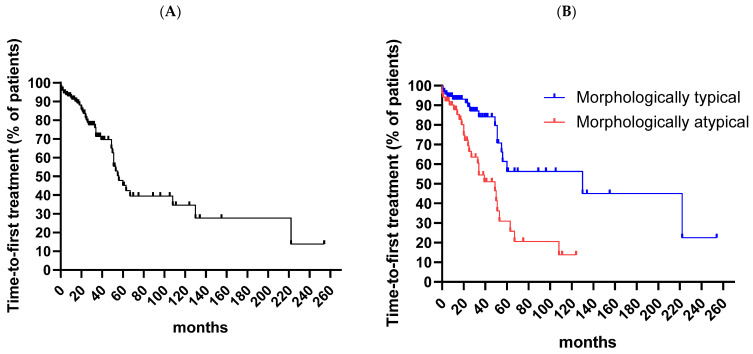
Time-to-first treatment (TTFT). (**A**) The median time-to-first treatment (TTFT) for the whole cohort was 56 months. (**B**) The median time-to-first treatment (TTFT) was 130 months and 49 months for the morphologically typical and atypical cohorts, respectively (*p* = 0.0004). (**C**) The median time-to-first treatment (TTFT) was 130 months and 49 months for the morphologically typical and atypical cohorts, respectively (*p* = 0.0004). (**D**) The median time-to-first treatment (TTFT) was 63 months and 49 months for the immunophenotypically typical and atypical cohorts, respectively (ns).

**Table 1 cancers-16-00469-t001:** Established morphological criteria used to classify CLL [3].

Classification	Typical	Atypical
Criteria	>90% of lymphocytes are small-to-medium sized with relatively normal morphology, except for a characteristically clumped, chunky chromatin pattern. Prolymphocytes and/or large cells < 10% of circulating lymphocytes	Small lymphocytes plus >10% and <55% prolymphocytes; Mixed-cell subtype: >15% lymphoplasmacytoid cells, cells exhibiting nuclear indentations/clefts, or both with; prolymphocytes < 10% of cells

**Table 2 cancers-16-00469-t002:** Types of leukemic B lymphoid cells according to FAB classification [3].

Cell Type (Disease)	Size	Chromatin	Nucleolus	Cytoplasm	Other Features
Small lymphocytes (CLL)	<2 red blood cells	Clumped in coarse blocks	Absent	Scanty high nuclear: cytoplasmic ratio	Regular nuclear outline
Large lymphocytes (CLL, mixed cell)	>2 red blood cells	Clumped	Inconspicuous or small	Low nuclear: cytoplasmic ratio, variable	Variable size
Prolymphocytes (PLL)	>2 red blood cells	Clumped	One, prominent	Low nuclear: cytoplasmic ratio	Variable size
Pleomorphic prolymphocytes (CLL/PL)	>2 red blood cells	Clumped	Central and prominent	Variable nuclear: cytoplasmic ratio	Variable size
Cleft cells (FL)	1–2 red blood cells	Homogeneously coarse	Absent or one or two inconspicuous	Scanty; not visible or narrow rim	One or two shallow or deep narrow nuclear clefts from angular base

**Table 3 cancers-16-00469-t003:** Demographics and CLL biological characteristics of patients at diagnosis.

Parameters	
Age, median (range)	67 (38–90) (*n* = 153)
Males, number (%)	94 (61%) (*n* = 153)
Rai stage, number (%)	(*n* = 152)
0	82 (54%)
I	25 (10%)
II	45 (30%)
III	0
IV	10 (7%)
Binet stage, number (%)	(*n* = 152)
A	93 (61%)
B	49 (32%)
C	10 (7%)
White blood cell count, ×10^9^/L, median (range)	17.7 (1.2–230) (*n* = 151)
Lymphocyte count, ×10^9^/L, median (range)	12.4 (4.3–200) (*n* = 151)
Hemoglobin, g/dL, median (range)	13.8 (8–16.8) (*n* = 149)
Platelet count, ×10^9^/L, median (range)	177 (33–462) (*n* = 149)
Beta2-microglobulin, mg/L, median (range)	2.5 (1.4–10.9) (*n* = 96)
Lactate dehydrogenase, UI/L, median (range)	186 (126–909) (*n* = 113)
IGHV unmutated, number (%)	50 (40%) (*n* = 125)
FISH abnormalities, number (%) *	(*n* = 128)
Negative	33 (26%)
Deletion 13q	63 (49%)
Trisomy 12	21 (16%)
Deletion 11q	8 (6%)
Deletion 17p	(2%)
CD5 positive, number (%)	153 (100%) (*n* = 153)
CD23 positive, number (%)	148 (97%) (*n* = 153)
FMC7 positive, number (%)	34 (25%) (*n* = 135)
CD79b positive, number (%)	81 (54%) (*n* = 150)
CD200 positive, number (%)	146 (100%) (*n* = 146)
CD20 expression, number (%)	(*n* = 151)
low	89 (59%)
intermediate	41 (27%)
high	21 (14%)
Surface immunoglobulin light chain intensity, number (%)	(*n* = 150)
low	90 (60%)
intermediate	46 (31%)
high	14 (9%)
CD43 positive, number (%)	141 (93%) (*n* = 151)
CD38 positive, number (%)	53 (48%) (*n* = 111)
CD49d positive, number (%)	62 (48%) (*n* = 129)

* Grouped according to Dohner’s hierarchical classification [12].

**Table 4 cancers-16-00469-t004:** Immunologic markers and their significance.

Monoclonal Antibody	Significance	References
CD5	Thymocytes, mature T-cells, subpopulations of B cells	[18,19]
CD20	Subpopulations of precursor B cells, B cells	[20,21]
CD22	Surface expression in mature B cells, cytoplasmic expression in precursor B cells	[20,22]
CD23	Subpopulations of B cells	[23,24]
CD38	Most thymocytes, activated mature T lymphocytes, B lymphocyte precursors, germinal center B cells, plasma cells	[25,26]
CD43	T cells, natural killer (NK) cells, pre-B, and activated B cells, granulocytes	[9,27]
CD49d	T and B lymphocytes and weakly on monocytes	[28,29]
CD79b	Surface of Ig-positive B cells and cytoplasm of Ig-negative B cell precursors	[6,30]
CD200	Thymocytes, CD19+ B cells, subpopulations of T cells	[7,31,32]
FMC7	Differentiated B lymphocytes	[33,34]
Kappa–Lambda light chain	Surface of mature B cells	[33,35]

**Table 5 cancers-16-00469-t005:** Scoring system adopted for defining immunophenotypic atypical CLL.

MoAb	Score
CD20 high density	1
SmIg high density	1
FMC7 expression	1
CD79b expression	1
CD200 negativity	1

**Table 6 cancers-16-00469-t006:** Demographics and CLL biological characteristics of patients at diagnosis, categorized based on morphological classification.

Parameters	Morphologically Typical CLL	Morphologically Atypical CLL	
Age, median (range)	67 (40–90) (*n* = 97)	67 (38–89) (*n* = 56)	NS
Males, number (%)	58 (60%) (*n* = 97)	36 (64%) (*n* = 56)	NS
Rai stage, number (%)	(*n* = 97)	(*n* = 55)	NS
0–I	66 (68%)	31 (56%)	
II–IV	31 (32%)	24 (44%)	
Binet stage, number (%)	(*n* = 97)	(*n* = 55)	NS
A	60 (62%)	33 (60%)	
B	32 (33%)	17 (31%)	
C	5 (5%)	5 (9%)	
White blood cell count, ×10^9^/L, median (range)	17.4 (8.75–230) (*n* = 97)	18.64 (1.2–93) (*n* = 54)	NS
Lymphocyte count, ×10^9^/L, median (range)	11.46 (5.1–200) (*n* = 97)	13.4 (4.3–86.6) (*n* = 54)	NS
Hemoglobin, g/dL, median (range)	13.7 (8–16.8) (*n* = 97)	13.8 (10.7–16.6) (*n* = 52)	NS
Platelet count, ×10^9^/L, median (range)	175 (33–462) (*n* = 97)	182 (45–337) (*n* = 52)	NS
Beta2-microglobulin, mg/L, median (range)	2.4 (1.4–5.6) (*n* = 61)	2.5 (1.5–10.9) (*n* = 35)	NS
Lactate dehydrogenase, UI/L, median (range)	184 (126–530) (*n* = 68)	190 (139–909) (*n* = 45)	NS
IGHV unmutated, number (%)	27 (33%) (*n* = 82)	23 (53%) (*n* = 43)	*p* = 0.0258
FISH abnormalities, number (%) *	(*n* = 83)	(*n* = 45)	NS
Negative	24 (29%)	9 (20%)	
Deletion 13q	42 (51%)	21 (47%)	
Trisomy 12	10 (12%)	11 (24%)	
Deletion 11q	4 (5%)	4 (9%)	
Deletion 17p	3 (4%)	0	
CD5 positive, number (%)	97 (100%) (*n* = 97)	56 (100%) (*n* = 56)	NS
CD23 positive, number (%)	96 (99%) (*n* = 97)	52 (93%) (*n* = 56)	NS
FMC7 positive, number (%)	21 (23%) (*n* = 93)	13 (31%) (*n* = 42)	NS
CD79b positive, number (%)	48 (50%) (*n* = 96)	33 (61%) (*n* = 54)	NS
CD200 positive, number (%)	94 (100%) (*n* = 94)	52 (100%) (*n* = 52)	NS
CD20 expression, number (%)	(*n* = 97)	(*n* = 54)	*p* < 0.0001
low	72 (74%)	17 (31%)	
intermediate	18 (19%)	23 (43%)	
high	7 (7%)	14 (26%)	
Surface immunoglobulin light chain intensity, number (%)	(*n* = 96)	(*n* = 54)	NS
low	63 (66%)	27 (50%)	
intermediate	24 (25%)	22 (41%)	
high	9 (9%)	9 (9%)	
CD43 positive, number (%)	92 (95%) (*n* = 97)	49 (91%) (*n* = 54)	NS
CD38 positive, number (%)	26 (38%) (*n* = 71)	26 (65%) (*n* = 40)	*p* < 0.0063
CD49d positive, number (%)	38 (44%) (*n* = 87)	24 (57%) (*n* = 42)	NS

* Grouped according to Dohner’s hierarchical classification [12].

**Table 7 cancers-16-00469-t007:** Demographics and CLL biological characteristics of patients at diagnosis, categorized based on immunophenotypic classification.

Parameters	Immunophenotypically Typical CLL	Immunophenotypically Atypical CLL	
Age, median (range)	67 (38–90) (*n* = 117)	69 (47–88) (*n* = 36)	NS
Males, number (%)	72 (62%) (*n* = 117)	22 (61%) (*n* = 36)	NS
Rai stage, number (%)	(*n* = 117)	(*n* = 35)	*p* = 0.0111
0–I	81 (69%)	16 (46%)	
II–IV	36 (31%)	19 (53%)	
Binet stage, number (%)	(*n* = 117)	(*n* = 35)	*p* = 0.0208
A	78 (67%)	15 (43%)	
B	31 (26%)	18 (51%)	
C	8 (7%)	2 (6%)	
White blood cell count, ×10^9^/L, median (range)	17.9 (8.2–230) (*n* = 116)	15.9 (1.2–202) (*n* = 35)	NS
Lymphocyte count, ×10^9^/L, median (range)	12.83 (4.9–200) (*n* = 116)	10.7(4.3–139) (*n* = 35)	NS
Hemoglobin, g/dL, median (range)	13.8 (8–16.8) (*n* = 114)	13.5 (9.6–16.6) (*n* = 35)	NS
Platelet count, ×10^9^/L, median (range)	177 (33–462) (*n* = 114)	174 (86–304) (*n* = 35)	NS
Beta2-microglobulin, mg/L, median (range)	2.4 (1.4–10.9) (*n* = 72)	2.6 (1.5-5.3) (*n* = 24)	NS
Lactate dehydrogenase, UI/L, median (range)	185 (126–909) (*n* = 88)	188 (149–607) (*n* = 25)	NS
IGHV unmutated, number (%)	41 (43%) (*n* = 96)	9 (31%) (*n* = 29)	NS
FISH abnormalities, number (%) *	(*n* = 99)	(*n* = 29)	NS
Negative	28 (28%)	5 (17%)	
Deletion 13q	50 (51%)	13 (45%)	
Trisomy 12	13 (13%)	8 (28%)	
Deletion 11q	6 (6%)	2 (7%)	
Deletion 17p	2 (2%)	1 (3%)	
CD5 positive, number (%)	117 (100%) (*n* = 117)	36 (100%) (*n* = 36)	NS
CD23 positive, number (%)	113 (97%) (*n* = 117)	35 (97%) (*n* = 36)	NS
FMC7 positive, number (%)	15 (14%) (*n* = 107)	19 (68%) (*n* = 28)	*p* < 0.0001
CD79b positive, number (%)	49 (43%) (*n* = 114)	32 (89%) (*n* = 36)	*p* < 0.0001
CD200 positive, number (%)	110 (100%) (*n* = 110)	36 (100%) (*n* = 36)	NS
CD20 expression, number (%)	(*n* = 115)	(*n* = 36)	*p* < 0.0001
low	77 (67%)	12 (33%)	
intermediate	33 (29%)	8 (22%)	
high	5 (4%)	16 (44%)	
Surface immunoglobulin light chain intensity, number (%)	(*n* = 115)	(*n* = 35)	*p* < 0.0001
low	79 (69%)	11 (31%)	
intermediate	31 (27%)	15 (43%)	
high	5 (4%)	9 (26%)	
CD43 positive, number (%)	113 (98%) (*n* = 115)	28 (78%) (*n* = 36)	*p* < 0.0001
CD38 positive, number (%)	35 (42%) (*n* = 83)	18 (64%) (*n* = 28)	NS
CD49d positive, number (%)	38 (39%) (*n* = 98)	24 (77%) (*n* = 31)	*p* = 0.0002

* Grouped according to Dohner’s hierarchical classification [12].

**Table 8 cancers-16-00469-t008:** Concordance between categorizations.

Parameters	Immunophenotypically and Morphologically Typical CLL 74/153 (48%)	Immunophenotypically and Morphologically Atypical CLL 13/153 (8%)	Discordant: Immunophenotypically Typical/Morphologically Atypical CLL 43/153 (28%)	Discordant: Morphologically Typical/Immunophenotypically Atypical CLL 23/153 (15%)
Age, median (range)	67 (40–90) (*n* = 74)	68 (47–88) (*n* = 13)	67 (38–89) (*n* = 43)	70 (50–83) (*n* = 23)
Males, number (%)	43 (58%) (*n* = 74)	7 (54%) (*n* = 13)	29 (67%) (*n* = 43)	15 (65%) (*n* = 23)
Rai stage, number (%)	(*n* = 74)	(*n* = 12)	(*n* = 23)	(*n* = 12)
0–I	56 (76%)	6 (50%)	10 (58%)	6 (43%)
II–IV	18 (24%)	6 (50%)	13 (42%)	6 (57%)
Binet stage, number (%)	(*n* = 74)	(*n* = 12)	(*n* = 43)	(*n* = 23)
A	51 (69%)	6 (50%)	27 (63%)	9 (39%)
B	18 (24%)	4 (33%)	13 (30%)	14 (61%)
C	5 (7%)	2 (17%)	3 (7%)	0
White blood cell count, ×10^9^/L, median (range)	17.9 (10–230) (*n* = 74)	19.9 (1.2–32.6) (*n* = 12)	18.3 (8.2–93) (*n* = 42)	15.5 (8.7–202) (*n* = 23)
Lymphocyte count, ×10^9^/L, median (range)	12.75 (5.1–200) (*n* = 74)	15.6 (4.3–21.9) (*n* = 12)	12.8 (8.9–86.6) (*n* = 42)	10.3 (5.3–139) (*n* = 23)
Hemoglobin, g/dL, median (range)	13.8 (8–16.8) (*n* = 74)	13.7 (10.7–16.6) (*n* = 12)	13.9 (10.7–16.6) (*n* = 40)	13.2 (9.6–15.5) (*n* = 23)
Platelet count, ×10^9^/L, median (range)	175 (33–462) (*n* = 74)	169 (86–304) (*n* = 12)	191 (45–337) (*n* = 40)	174 (102–265) (*n* = 23)
Beta2-microglobulin, mg/L, median (range)	2.4 (1.4–5.6) (*n* = 47)	2.6 (1.5–4.9) (*n* = 10)	2.5 (1.7–10.9) (*n* = 25)	2.6 (1.6–5.3) (*n* = 14)
Lactate dehydrogenase, UI/L, median (range)	183 (126–530) (*n* = 54)	181 (149–607) (*n* = 11)	195 (139–909) (*n* = 34)	188 (153–268) (*n* = 14)
IGHV unmutated, number (%)	20 (31%) (*n* = 64)	2 (18%) (*n* = 11)	21 (66%) (*n* = 32)	7 (39%) (*n* = 18)
FISH abnormalities, number (%)	(*n* = 65)	(*n* = 11)	(*n* = 34)	(*n* = 18)
Negative	20 (31%)	1 (9%)	8 (24%)	4 (22%)
Deletion 13q	34 (52%)	5 (45%)	16 (47%)	8 (44%)
Trisomy 12	7 (11%)	5 (45%)	6 (18%)	3 (17%)
Deletion 11q	2 (3%)	0	4 (12%)	2 (11%)
Deletion 17p	2 (3%)	0	0	1 (6%)
CD5 positive, number (%)	74 (100%) (*n* = 74)	13 (100%) (*n* = 13)	43 (100%) (*n* = 43)	23 (100%) (*n* = 23)
CD23 positive, number (%)	73 (99%) (*n* = 74)	12 (92%) (*n* = 13)	40 (93%) (*n* = 43)	23 (100%) (*n* = 23)
FMC7 positive, number (%)	7 (10%) (*n* = 73)	5 (63%) (*n* = 8)	8 (24%) (*n* = 34)	14 (70%) (*n* = 20)
CD79b positive, number (%)	28 (38%) (*n* = 73)	12 (92%) (*n* = 13)	21 (51%) (*n* = 41)	20 (87%) (*n* = 23)
CD200 positive, number (%)	71 (100%) (*n* = 71)	13 (100%) (*n* = 13)	39 (100%) (*n* = 39)	23 (100%)(*n* = 23)
CD20 expression, number (%)	(*n* = 74)	(*n* = 13)	(*n* = 41)	(*n* = 23)
low	60 (81%)	0	17 (41%)	12 (52%)
intermediate	13 (18%)	3 (23%)	20 (49%)	5 (22%)
high	1 (1%)	10 (77%)	4 (10%)	6 (26%)
Surface immunoglobulin light chain intensity, number (%)	(*n* = 73)	(*n* = 12)	(*n* = 42)	(*n* = 23)
low	53 (73%)	1 (8%)	26 (62%)	10 (43%)
intermediate	17 (23%)	8 (67%)	14 (33%)	7 (30%)
high	3 (4%)	3 (25%)	2 (5%)	6 (26%)
CD43 positive, number (%)	73 (99%) (*n* = 74)	9 (69%) (*n* = 13)	40 (98%) (*n* = 41)	19 (83%) (*n* = 23)
CD38 positive, number (%)	17 (33%) (*n* = 52)	8 (89%) (*n* = 9)	18 (58%) (*n* = 31)	10 (53%) (*n* = 19)
CD49d positive, number (%)	25 (37%) (*n* = 67)	11 (100%) (*n* = 11)	13 (42%) (*n* = 31)	13 (65%) (*n* = 20)

**Table 9 cancers-16-00469-t009:** Most relevant published studies investigating the frequency and prognostic significance of atypical CLL.

Reference	No. of Evaluated Patients	Criteria for Defining Atypical CLL	Atypical CLL (No. and %)	Impact on Prognosis (Correlations)
Matutes E et al., 1994 [5]	400	FMC	52/400 (13%)	
Finn et al., 1996 [10]	26	Morphology FCM	10/26 (38%) 8/26 (31%)	Trisomy 12 (*p* 0.004) and atypical immunophenotype (*p* 0.13) despite this latter having no statistical significance
Criel A et al., 1997 [14]	390	Morphology	90/390 (23.1%)	Aberrant immunophenotype in 33% of cases: FMC7 positivity (*p* < 0.0001), intensive smIg (*p* < 0.0001). Trisomy 12 in 36% of cases (*p* < 0.0001); del11q the second most common anomaly (13.5%). More frequent advanced clinical stage (*p* < 0.05), lymph node involvement (*p* < 0.05), time-to-treatment shorter (*p* < 0.05), shorter survival (*p* < 0.005)
D’Arena G et al., 2001 [15]	84	Morphology	15 (18%)	Higher expression of CD20 and CD22, CD79b and FMC7 expression and smIg density
Schwarz et al., 2006 [16]	88	Morphology	63/88 (71.6%)	Inferior OS (103 vs. 237 months; p 0.03), unmutated IgVH (81.8%), trisomy 12 and del17p, CD38+ slightly more frequent (p ns)
Habib LK and Finn WG, 2006 [17]	81	FCM	14/81 (17%)	High CD20, CD22, FMC7, and smIg, and low CD23
Marianneaux et al., 2013 [36]	97	Morphology	26/97 (27%)	Higher prevalence of trisomy 12, unmutated IgVH, CD38 expression, lower prevalence of del13q14, higher fluorescence expression of CD79b
Ting et al., 2018 [11]	63	FMC *	7/63 (11%)	Not reported

FCM: flow cytometry; * Matutes score < 3 (without molecular, cytogenetics and immunohistochemistry evidence of mantle cell lymphoma).

## Data Availability

The data are not publicly available due to participant identifiability. The datasets used and/or analyzed during the current study are available from the corresponding author on reasonable request.

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
