# Peer review of "What Does Atypical Chronic Lymphocytic Leukemia Really Mean? A Retrospective Morphological and Immunophenotypic Study"

_cancers, 2024, doi:10.3390/cancers16020469_

Round 1

Reviewer 1 Report

Comments and Suggestions for Authors

It is a good review type study. The authors should address the followings:

1. It is not clear to me what the final conclusion it.

2. There are so many markers mentioned; all readers cannot recall what their significance is for B cells and B cell-related malignancy. I suggest making a table (could be supplementary one) and explain all of them with pertinent references.

Comments on the Quality of English Language

In general English is OK, however, the take home message is not clear

Author Response

  • Reviewer no. 1
  1. The final conclusions have been modified to be better understood.
  2. A specific Table has been added (no. 4) with pertinent references.

Reviewer 2 Report

Comments and Suggestions for Authors

Atypical CLL is a poor defined CLL variant, usually identify by morphology. In the present paper the authors performed a retrospective analysis of the CLL cases referred to their Institution from 2001 o 2019, trying to associate morphology,  immunophenotype and molecular markers to a better diagnostic  and a possible prognostic  definition.

They concluded that morphological features identify a subgroup of patients with shorter time to first treatment and that atypical cases for immunophenotype and morphology have a “overall worse prognosis”.

They do not find differences in distribution of cytogenetic abnormalities among typical, atypical and immunophenotypically/morphologically discordant groups, suggesting that morphology is still superior to cytogenetics in providing prognostic information. 

 The paper is interesting and highlight the need of a major biological definition of CLL variants. 

I have just few question:

1.     In table 5 the authors report an higher proportion of cases diagnosed in advanced stages in the atypical group. How did they explain this fact?

2.     In the morphological atypical subgroup there are more unmutated IGVH cases? This may be expression of the development of a different cell ?

3.     They used TTFT as surrogate marker of prognosis, but it would be interesting to know if there was a different response to therapy between typical and atypical cases.

4.     Cytogenetic abnormalities usually associated with prognosis in CLL do not have different distribution/frequency by groups). How about different genes? Could a NGS analysis provide a better stratification? Pleas add a comment in “discussion”

Author Response

  • Reviewer no. 2
  1. In Results the following statement has been added: “In light of this, in the morphological atypical subgroup more advanced stages of patients were found”. This may be in line with the poor prognostic significance of this type of CLL.
  2. In Discussion the following statement has been added: “Whether the differences between morphologically typical and atypical cases in terms of IgVH mutational status and CD38 and CD20 expression might be an expression of a somehow different disease in terms of cell-of-origin unmutated IgVH cases more observed in atypical CLL could be expression of the development of a different cell needs to be addressed by specific studies”.
  3. The heterogeneity of therapies given does not permit a reliable analysis of their impact according to atypical and typical CLL entities.
  4. NGS were not performed in practical routine approach to CLL. For that reason we have no data according to.